# Galad Score as a Prognostic Marker for Patients with Hepatocellular Carcinoma

**DOI:** 10.3390/ijms242216485

**Published:** 2023-11-18

**Authors:** Silvia Cagnin, Rossella Donghia, Andrea Martini, Pasqua Letizia Pesole, Sergio Coletta, Endrit Shahini, Giulia Boninsegna, Alessandra Biasiolo, Patrizia Pontisso, Gianluigi Giannelli

**Affiliations:** 1Department of Medicine, University of Padova, 35123 Padova, Italy; silvia.cgn@gmail.com (S.C.); andrea.martini@aopd.veneto.it (A.M.); giulia.boninsegna@studenti.unipd.it (G.B.); alessandra.biasiolo@unipd.it (A.B.); patrizia@unipd.it (P.P.); 2National Institute of Gastroenterology–IRCCS “Saverio de Bellis”, 70013 Castellana Grotte, Italy; rossella.donghia@irccsdebellis.it (R.D.); letizia.pesole@irccsdebellis.it (P.L.P.); sergio.coletta@irccsdebellis.it (S.C.); endrit.shahini@irccsdebellis.it (E.S.)

**Keywords:** hepatocellular carcinoma, GALAD, prognosis, biomarker

## Abstract

Background: Hepatocellular carcinoma (HCC) accounts for more than 75% of primary liver cancers, which are the second leading cause of cancer-related deaths. The GALAD (gender, age, AFP-L3, AFP, and des-carboxy-prothrombin) score is a diagnostic tool developed based on gender, age, alpha-fetoprotein, alpha-fetoprotein L3, and des-gamma-carboxy prothrombin, originally designed as a diagnostic tool for HCC in high-risk patients. Methods: We analyzed 212 patients with and without cirrhosis. The population study was divided into patients with liver cirrhosis without evidence of HCC at the time of serum sample collection for GALAD score determination and patients with liver cirrhosis and a confirmed diagnosis of HCC at the time of serum sample collection for GALAD score determination. Patients were followed up until death or liver transplantation. The association between variables and HCC mortality risk was performed, and the results were presented as hazard ratio (HR). The receiver operating characteristic (ROC) curve was used to assess the performance of the GALAD HCC diagnosis. The survival probability was explored using the non-parametric test, and the equality of survival amongst categories was assessed with the log-rank test. Results: Biomarkers were higher in the HCC group compared to cirrhosis. Kaplan–Meier survival probability analysis for individual GALAD categories revealed that a high GALAD level was associated with decreased survival during follow-up, and the difference between the curves was statistically significant (*p* = 0.01). Conclusions: Our findings suggest that the GALAD score has promise as a prognostic tool, with implications for improving patient management and treatment strategies for HCC.

## 1. Introduction

Primary liver cancer is the seventh-most common type of cancer and the second-leading cause of cancer-related deaths, with hepatocellular carcinoma (HCC) accounting for more than 75% of liver tumors [1,2]. HCC incidence progressively increases with age, peaking at 70 years old and presenting a 2–3 times higher incidence and mortality in males [1,3,4]. Notably, while other cancers are showing a decreasing mortality, HCC is one of the fastest-growing causes of cancer-related death worldwide. The high mortality can be attributed to various factors, including an inadequate early detection strategy, risk stratification methods, a lack of curative treatment for advanced tumors and the added risk of mortality from impaired liver function [4,5,6]. Viral causes of HCC are decreasing due to the introduction of HBV (Hepatitis B virus) vaccination and curative therapies for HCV (Hepatitis C virus), but non-viral causes are on the rise (alcohol and metabolic-induced liver disease), especially in Western countries [2,4].

Patients with HCC have access to many therapeutic options, and the international guidelines (European association for the study of the liver and American association for the study of liver diseases) assign a treatment based on the Barcelona clinic liver cancer (BCLC) staging system. The overall 5-year survival in patients with HCC remains below 15%, except for those patients receiving potentially curative treatment (e.g., surgical resection, ablation, or liver transplantation), which are options available only for BCLC 0-A patients; these patients can achieve a 5-year survival rate ranging from 40 to 70% [7], while patients with a more advanced disease receiving systemic therapy including immune checkpoint inhibitors (ICIs) display even worse survival [8].

Recognizing the importance of early diagnosis in improving prognosis leads to the importance of developing effective screening strategies. The current guidelines endorse semi-annual screening for high-risk patients using abdominal ultrasound (US) with or without alfa-fetoprotein, as the combination of these two tools increases sensitivity from 45% of US alone to 63% [2]. However, this surveillance strategy has several limitations, including operator-dependent ultrasound results and frequent inconclusiveness in obese patients and those with non-alcoholic fatty liver disease [9].

To improve the current surveillance strategies, many biomarkers and scores are under investigation, with single biomarkers usually having suboptimal performance. The GALAD score is based on clinical and laboratory parameters, including gender, age, alpha-fetoprotein (AFP), lens culinaris agglutinin-reactive AFP (AFP-L3), and des-gamma-carboxy prothrombin (DCP). This score has proven useful in early HCC diagnosis [10,11], having a higher effectiveness and sensitivity compared to individual parameters for both early and advanced tumors, regardless of the underlying etiology. Particular interest in this score was raised in those patients with a suspected small tumor and negative AFP and DCP [12,13]. AFP is the sole biomarker, which is widely used for HCC early detection in patients with chronic liver disease under surveillance. Notably, AFP is currently the only serum biomarker to have undergone all five phases of biomarker validation, and it is the only biomarker backed by high-quality supporting evidence, making it suitable for predicting prognosis as well [14]. AFP-L3is a fucosylated glycoform of AFP produced by HCC cells, and high levels can be found in patients with advanced liver disease [3,5]. DCP is an abnormal prothrombin variant produced by HCC cells, and its levels are usually undetectable in healthy individuals [15]. High DCP levels have been associated with advanced liver disease, especially portal vein invasion [3]. Furthermore, the GALAD score includes demographic variables (gender and age) that are associated with an elevated HCC risk and are readily accessible data. Although the GALAD score is a potentially efficient diagnostic tool, there is little research on its role as a prognostic tool.

The aim of our study was to move from the current utilization of the GALAD score as a screening tool for early disease to its possible application as a prognostic score.

## 2. Results

In our retrospective cohort, we observed 212 patients with cirrhosis, which were subdivided into 112 patients with HCC and 100 patients without HCC, as reported in Table 1.

Patients in the HCC group were significantly older than patients in the cirrhosis group (67.22 vs. 59.36, *p* < 0.0001), and there was a higher prevalence of males (75.89% vs. 67.00%), although not statistically significant. Patients in the HCC group had higher levels of gamma-glutamyl transferase (GGT), aspartate transferase (AST), and bilirubin, which were statistically significant (*p* = 0.0008, *p* < 0.0001, and *p* = 0.04, respectively). Biomarkers included in the GALAD score, i.e., AFP, AFP-L3, and DCP, were higher in the HCC group compared to the cirrhosis group with statistically significant *p*-values (*p* < 0.0001). We observed a strong association, with clearly different proportions between the two groups, regarding the different etiology in the two groups (*p* = 0.02). A higher prevalence of HCV and alcoholic etiologies was indeed found in the HCC group (45.05% and 36.04%, respectively), while HBV and metabolic etiologies were predominant in the control group (19.0% and 14.0%, respectively). The model for end-stage liver disease (MELD) score and the GALAD score had higher median values in the HCC group (11.00 vs. 10.00 with *p* = 0.05, and 1.38 vs. −2.68 with *p* < 0.0001, respectively). Additionally, when GALAD was used as a categorical variable, it had higher prevalence values and was more represented in the HCC group (81.00 vs. 12.00, *p* < 0.001). Table 2 displays the results of univariate Cox regression models used to test the associations of variables in HCC patients with mortality risk.

Blood parameters such as AST, bilirubin, creatinine, INR, AFP, AFP-L3, and DCP exhibited positive associations with the highest risk of mortality (all with *p* < 0.005). The same trend was observed for Child–Pugh stage C, MELD scores, and BCLC stage B, all of which were strongly associated with an increase in mortality (HR = 3.930, *p* < 0.001, 1.812 to 8.164 95% C.I.; HR = 1.136, *p* = 0.006, 95% C.I. 1.036 to 1.245; and HR = 2.188, *p* = 0.028, 95% C.I. 1.086 to 4.407, respectively). Both continuous and categorical forms of the GALAD score were associated with a higher risk of mortality (HR = 1.194 with *p* < 0.0001, and HR = 2.704 with 0.020 for patients included in the category of ≥−0.95). A set of a priori predetermined variables (AST, MELD, BCLC, GALAD, and treatment) was selected to be included together in a final multivariate model (Table 3), and a stepwise backward method was applied. Variables already incorporated into the scoring systems (e.g., bilirubin and creatinine) were excluded from the statistical analysis.

To identify the best cut-off for the GALAD score, we constructed an ROC curve that showed a high area under curve (AUC) (0.8862). Using the cut-off value of −0.95, the GALAD score demonstrated a strong performance in detecting HCC, with a sensitivity of 80.41%, a specificity of 87.76%, a positive predictive value (PPV) of 86.67%, and a negative predictive value (NPV) of 81.90% (Figure 1a). This performance was better than that obtained with the other individual biomarkers, showing lower AUC values (Figure 1b–d).

Kaplan–Meier survival probability analyses for individual GALAD categories revealed that a GALAD level > −0.95 was associated with decreased survival compared to lower levels of this biomarker (Figure 2a), and the difference between the curves was statistically significant (*p* = 0.01).

In the competing risk analysis (Figure 2b), considering liver transplantation as a competing event, the levels of a GALAD score > −0.95 were significantly associated with lower survival probability (sub-distribution hazard ratios (SHR) = 3.19, *p* = 0.008, 1.36 to 1.7. 47 95% C.I.).

## 3. Discussion

Inadequate early HCC detection under surveillance and risk stratification methods are two critical factors that can contribute to an increase in HCC fatalities in high-risk populations, highlighting the need for more accurate surveillance tests. To date, AFP and US are the only diagnostic tests used for HCC detection in cirrhotic patients undergoing surveillance [3]. Moreover, we do not have efficient, non-invasive prognostic tools to better stratify these patients.

The GALAD score is a diagnostic model that was initially developed using data from 394 patients with HCC and 439 patients with chronic liver disease in the United Kingdom (UK) [10]. In this cohort, the GALAD score demonstrated high overall performance in the early detection of HCC (AuROC of 92%, sensitivity of 92%, and specificity of 79%) [10]. The GALAD model was then validated in independent cohorts of 6834 mixed Asian and European patients, preserving high diagnostic accuracy across the populations and proving to be as effective in detecting early-stage HCCs as late-stage HCCs [16]. Further studies [9,17] confirmed that the GALAD algorithm outperformed the combination of the individual biomarkers. Interestingly, in two American phase II and phase III studies conducted in 2019, the AuROC of GALAD for the detection of HCC was higher than that of the US, both of which were outperformed by their combination (GALADUS) [18].

To the best of our knowledge, the prognostic ability of the GALAD score has not yet been extensively studied. In a cross-sectional and longitudinal multicenter case–control study of 1.561 Chinese patients, the GALAD score could detect the development of HCC 6 or 12 months before clinical diagnosis (AuROCs of 85% and 83%, respectively) [19]. Moreover, GALAD demonstrated the highest sensitivity for early HCC detection (53.8%, 63.3%, and 61.8% within 6, 12, and 24 months) in a 2022 US prospective cohort phase III biomarker study including 534 patients, but increased false positive results from 21.5% to 22.9% [7].

In our study, AFP, AFP-L3, and DCP included in the GALAD score were significantly higher in the HCC group than in the cirrhosis control group. In agreement with previous studies, GALAD showed a good diagnostic ability for HCC detection (AUC: 0.8862; sensitivity: 80.41%; and specificity: 87.76%) at the cut-off of −0.95. When the time variable was included in the analysis, the GALAD score was the only variable associated with a higher mortality risk (HR = 1.285). To date, the literature is still conflicting, as in other recent but small phase III US cohorts, the GALAD score performed worse, with one demonstrating a sensitivity of 53.8% and another demonstrating a sensitivity of 30.8% [7,9].

In a 2016 Italian study, the levels of the biomarkers that make up GALAD were measured but in a small cohort of patients (44 patients with chronic liver disease (CLD) without HCC and 54 patients with HCC with HCV or HBV etiology). They demonstrated that serum biomarker levels were significantly higher in HCC patients compared to CLD patients, with an AuROC of 88%, 87%, and 87%, respectively. Furthermore, the combination of AFP, AFP-L3, and DCP outperformed a single biomarker in the detection of HCC [17] and also the competing risk analysis, which essentially confirmed the results of the Cox regression analysis, as GALAD outperformed in HCC detection. This performance was confirmed in a more recent study based on a larger cohort of 545 consecutive patients with compensated advanced chronic liver disease without suspected focal lesions, followed every 6 months for up to 12 years, using liver imaging. The authors studied the performance of GALAD not only on the total cohort but also stratified on the viral, nonalcoholic fatty liver disease (NASH), and alcoholic sub-cohorts, highlighting a better performance for the alcohol sub-cohort. In contrast, GALAD did not work significantly in the NASH cohort, probably due to the different molecular mechanisms during the carcinogenesis process [20].

Serum biomarkers (AST, AFP, AFP-L3, and DCP) and Child–Pugh stage C and BCLC stage B were all positively associated with higher risks of mortality in HCC patients. When compared to the single biomarkers, the GALAD score performed better in detecting HCC, proving to be more effective than AFP, which is the most widely used biomarker (AuROC of 0.717 vs. 0.88, respectively).

Regarding survival probability, a higher GALAD score was significantly associated with decreased survival probability, even after the use of liver transplantation as a competing risk. A still unmet need is the use of serum biomarkers, including the GALAD score, to assess the response to therapy, in particular for those receiving drug-based therapy as ICIs.

Our study has some limitations due to its retrospective design, the limited number of patients included in the study, and the lack of a validation cohort. Nevertheless, this cohort of patients is well characterized by a long follow-up. Moreover, in our population, there is a higher prevalence of HCV and a lower prevalence of patients with NASH, even though these patients presented a high prevalence of metabolic comorbidities (40% of patients had arterial hypertension and 29% of them had diabetes), recalling the clinical characteristics of patients recently defined with metabolic dysfunction-associated steatotic liver disease or metabolic-associated fatty liver disease [21].

## 4. Materials and Methods

### 4.1. Patients

The study is based on the analysis of all medical records of eligible Italian outpatients with liver cirrhosis and with liver cirrhosis plus HCC who were prospectively and consecutively followed in the care management program [22] at the university hospital of Padova, as previously reported [23]. The median follow-up period was 14.50 months, with a range of 0–131 months.

The study, which was conducted according to the declaration of Helsinki, was part of a national project (FIRB Prot. RBLA03S4SP 005) and was approved by the local ethics committee. All patients provided written informed consent.

The population study was divided into two subclasses: (a) patients with liver cirrhosis without evidence of HCC at the time of serum sample collection for GALAD score determination, and (b) patients with liver cirrhosis and a confirmed diagnosis of HCC at the time of serum sample collection for GALAD score determination. Patients were followed up until death or liver transplantation. HCC diagnosis, based on either histology or on the presence of typical radiological features on one or more imaging techniques, followed national and international guidelines [3].

### 4.2. GALAD Score Determination

The immunoanalyzer TASWakoTM i30, micro total analysis system (FUJIFILM Wako Pure Chemical Corporation, Chuo-Ku, Osaka, Japan) was used to determine the serum levels of AFP, AFP-L3, and DCP. This automated system quantitatively measures the biomarker concentrations by microfluidic electrophoretic separation. The reportable range for AFP concentration is 0.3–1000 ng/mL. The reportable range for AFP-L3% is 0.5–99.5%. AFP-L3% is calculated as follows: AFP-L3% = AFP-L3 concentration/(AFP-L1 concentration + AFP-L3 concentration) × 100 [10]. The limit of detection for AFP-L1 and AFP-L3 was determined to be 0.030 ng/mL and 0.028 ng/mL, respectively. The reportable range for DCP concentration is 0.1–950 ng/mL, with expected values typically below 7.5 ng/mL. The limit of detection for DCP was determined to be 0.042 ng/mL.

The combination of these three biomarkers, with the addition of age and gender variables, produces the GALAD score. The formula is as follows: −10.08 + 1.67 × gender (1 for males, 0 for females) + 0.09 × age + 0.04 × AFP-L3% + 2.34 × log_10_AFP + 1.33 × log_10_DCP.

### 4.3. Statistical Analysis

Patients’ characteristics are presented as median interquartile range for continuous variables, and as frequency and percentage (%) for categorical variables. To test the association between the independent groups (HCC vs. cirrhosis), chi-square or Fisher test were employed for categorical variables as needed, while the Wilcoxon rank Mann–Whitney test was used for continuous variables.

The association between variables and the risk of HCC mortality risk was assessed using Cox’s proportional hazards regression model, and the results were presented as HR with a 95% confidence interval (95% C.I.). Multivariable analysis was conducted using the Cox proportional hazards model with stepwise backward method and a retention criterion of *p* < 0.10. This approach also allowed us to estimate the association of each variable independently, mitigating the effects of mutual collinearity. When scores for liver disease were included in the multivariate analysis, their components were excluded to avoid multicollinearity.

The AuROC curve was used to assess the performance of GALAD and other GALAD biomarkers in HCC diagnosis.

The cut-off value for GALAD was determined using the AuROC for HCC status assessment. Youden’s J statistic, which provides a single statistic capturing the performance of a dichotomous diagnostic test, was employed in conjunction with AuROC analysis. Sensitivity, specificity, PPV, and NPV were calculated accordingly.

Survival probability was explored using the non-parametric Kaplan–Meier method, and the equality of survival amongst categories was assessed with the log-rank test. For survival analysis, considering liver transplantation as a competing event, competing risk analysis was conducted, and SHR with a 95% C.I. was reported using the Fine and Gray method.

To test the null hypothesis of non-association, a two-tailed probability level of 0.05 was set. All analyses were performed using StataCorp. 2023. Stata Statistical Software: release 18. StataCorp LLC, College Station, TX, USA.

## 5. Conclusions

In conclusion, our study demonstrates that patients with HCC and a BCLC early stage with higher levels of GALAD scores had a lower probability of survival. The GALAD score is a potentially useful non-invasive prognostic tool for this cohort of patients, although further prospective and larger studies are needed to validate our results and investigate its prognostic use in other cohorts.

## Figures and Tables

**Figure 1 ijms-24-16485-f001:**
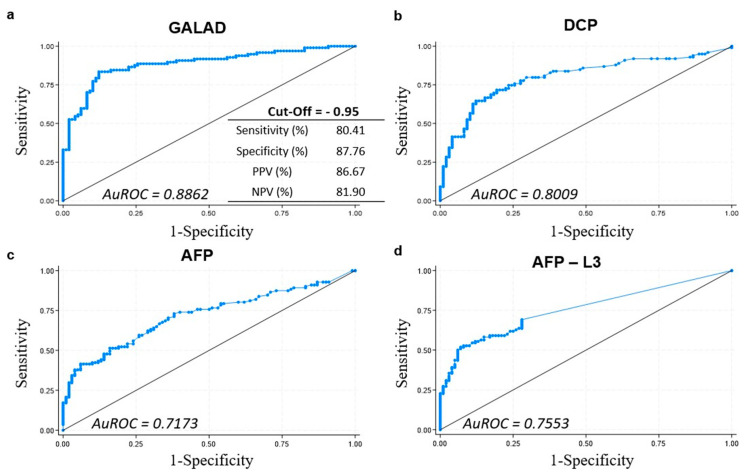
Area under the receiver operating characteristic (AuROC) curve of GALAD (**a**) and its components, DCP (**b**), AFP (**c**), and AFP-L3 (**d**), respectively.

**Figure 2 ijms-24-16485-f002:**
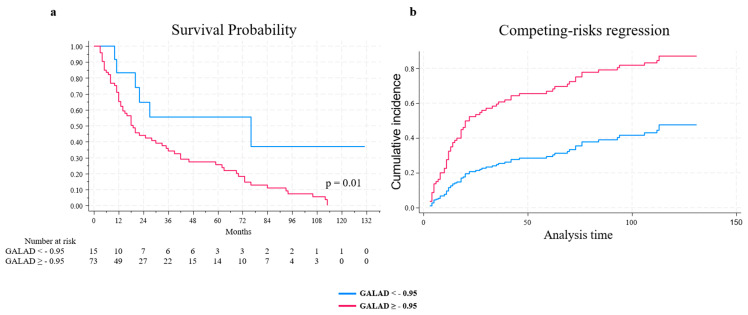
Kaplan–Meier curve comparing GALAD groups (**a**). Cumulative incidence with transplant as competing risk, stratified by GALAD categories (**b**).

**Table 1 ijms-24-16485-t001:** Comparison of blood and clinical parameters between HCC versus cirrhosis status.

Parameters *	Cirrhosis (*n* = 100)	HCC (*n* = 112)	*p* ^^^
Age (yrs)	59.36 (52.03–65.81)	67.22 (61.05–73.92)	<0.0001
Gender (M) (%)	67 (67.00)	85 (75.89)	0.15 ^Ψ^
BMI (Kg/m^2^)	25.35 (23.87–29.02)	25.95 (24.07–28.63)	0.71
GGT (U/L)	36.00 (29.00–87.00)	67.00 (40.00–111.00)	0.0008
AST (U/L)	38.00 (27.00–65.00)	63.00 (46.00–98.00)	<0.0001
ALT (U/L)	32.00 (26.00–56.00)	39.00 (25.00–71.00)	0.15
Bilirubin (mg/dL)	1.07 (0.57–1.97)	1.32 (0.86–2.13)	0.04
Creatinine (µg/dL)	0.84 (0.71–0.90)	0.84 (0.75–1.02)	0.07
INR	1.19 (1.08–1.38)	1.20 (1.12–1.34)	0.62
Albumin (g/L)	39.00 (33.30–42.00)	31.00 (26.00–36.00)	<0.0001
Platelets (10 × mm^3^)	120.00 (71.00–154.50)	98.00 (69.00–153.00)	0.37
AFP (ng/mL)	3.35 (2.00–6.80)	8.80 (3.60–48.70)	<0.0001
AFP-L3 (%)	0.29 (0.29–4.6)	7.75 (0.29–17.80)	<0.0001
DCP (ng/nL)	0.29 (0.21–0.50)	3.14 (0.49–23.51)	<0.0001
Diabetes (Yes) (%)	28 (28.00)	20 (29.41)	0.84 ^Ψ;^
Hypertension (Yes) (%)	35 (35.00)	27 (40.30)	0.49 ^Ψ;^
Etiology (%)			0.02 ^Ψ;^
Alcohol	31 (31.00)	40 (36.04)	
HBV	19 (19.00)	8 (7.21)	
HCV	33 (33.00)	50 (45.05)	
Metabolic	14 (14.00)	13 (11.71)	
Other	3 (3.00)	0 (0.00)	
Child–Pugh (%)			0.11 ^Ψ;^
A	63 (66.32)	52 (51.49)	
B	23 (24.21)	36 (35.64)	
C	9 (9.47)	13 (12.87)	
MELD	10.00 (9.00–13.00)	11.00 (8.00–12.00)	0.05
BCLC			--
0	--	18 (16.51)	
A	--	49 (44.95)	
B	--	42 (38.53)	
MTD (mm)	--	25.00 (17.50–40.00)	--
HCC Injury			--
Unifocality	--	51 (47.22)	
<3 Nodules	--	22 (20.37)	
Diffused	--	35 (32.41)	
Ascites (Yes) (%)	30 (30.03)	--	--
GALAD Score	−2.68 (−3.92–−1.72)	1.38 (−0.63–2.78)	<0.0001
GALAD Score (%)			<0.001 ^Ψ;^
<−0.95	86 (87.76)	16 (16.49)	
≥−0.95	12 (12.24)	81 (83.51)	
Treatment (Yes) (%)	--	76 (87.36)	--
Follow-Up (months)	--	14.50 (5.50–40.50)	--

* As median and interquartile range (IQR) for continuous variables and as frequency and percentage (%) for categorical. ^^^ Wilcoxon rank-sum test (Mann–Whitney). ^Ψ^ Chi-square or Fisher test where necessary. Abbreviations: AFP, alpha-fetoprotein; AST, aspartate transferase; ALT, alanine aminotransferase; BCLC, Barcelona clinic liver cancer; BMI, body mass index; DCP, des-gamma-carboxy prothrombin; GALAD, gender, age, AFP-L3, AFP and des-carboxy-prothrombin; GGT, gamma-glutamyl transferase; HBV, Hepatitis B virus; HCV, Hepatitis C virus; INR, international normalized ratio; MELD, model for end-stage liver disease; MTD, maximum tumor diameter.

**Table 2 ijms-24-16485-t002:** Cox regression model, on variable insert single in the model, to identify the rate of death among patients with HCC.

Parameters	HR	Se (HR)	*p*-Value	95% (C.I.)
Age	1.015	0.012	0.193	0.992 to 1.038
Gender (M)	0.661	0.172	0.113	0.396 to 1.103
BMI	0.981	0.030	0.519	0.924 to 1.041
GGT	1.001	0.001	0.143	0.999 to 1.003
AST	1.007	0.002	<0.001	1.003 to 1.010
ALT	1.003	0.003	0.278	0.997 to 1.008
Bilirubin	1.266	0.09	0.002	1.094 to 1.464
Creatinine	1.444	0.198	0.008	1.103 to 1.890
INR	7.531	5.822	0.009	1.655 to 34.271
AFP	1.000	0.00002	0.007	1.000 to 1.001
AFP-L3	1.018	0.005	<0.001	1.009 to 1.028
DCP	1.000	0.0001	0.003	1.000 to 1.001
Diabetes (Yes)	0.903	0.292	0.753	0.479 to 1.703
Hypertension (Yes)	1.072	0.342	0.826	0.573 to 2.006
Etiology (%)				
Alcohol (*Reference category*)	--	--	--	--
HBV	1.235	0.611	0.670	0.468 to 3.255
HCV	1.406	0.359	0.182	0.853 to 2.318
Metabolic	0.816	0.315	0.598	0.383 to 1.739
Child–Pugh				
A (*Reference category*)	--	--	--	--
B	1.492	0.397	0.133	0.885 to 2.514
C	3.930	1.466	<0.001	1.812 to 8.164
MELD	1.136	0.053	0.006	1.036 to 1.245
BCLC				
0 (*Reference category*)	--	--	--	--
A	1.346	0.464	0.388	0.685 to 2.646
B	2.188	0.782	0.028	1.086 to 4.407
MTD	1.014	0.005	0.005	1.004 to 1.023
HCC Injury				
Unifocality (*Reference category*)	--	--	--	--
<3 Nodules	1.001	0.333	0.998	0.521 to 1.921
Diffused	1.698	0.442	0.042	1.019 to 2.829
GALAD Score	1.194	0.044	<0.001	1.110 to 1.284
GALAD Score	2.330	0.572	0.001	1.440 to 3.769
<−0.95 (*Refeference category*)	--	--	--	--
≥−0.95	2.704	1.157	0.020	1.169 to 6.256
Treatment (Yes)	0.633	0.220	0.188	0.321 to 1.251

Abbreviations: AFP, alpha-fetoprotein; AST, aspartate transferase; ALT, alanine aminotransferase; BCLC, Barcelona clinic liver cancer; BMI, body mass index; DCP, des-gamma-carboxy prothrombin; GALAD, gender, age, AFP-L3, AFP, and des-carboxy-prothrombin; GGT, gamma-glutamyl transferase; HBV, Hepatitis B virus; HCV, Hepatitis C virus; HR, hazard ratio; INR, international normalized ratio; MELD, model for end-stage liver disease; MTD, maximum tumor diameter; se (HR), standard error of HR; 95% (C.I.), confidential interval at 95%.

**Table 3 ijms-24-16485-t003:** Cox regression model on parameters inserted together in the model to identify predictors of death among patients with HCC (A). Final multiple Cox model in stepwise method on variables included together in the model (B).

Parameters	HR	Se (HR)	*p*-Value	95% (C.I.)
**(A)**				
AST	1.009	0.006	0.137	0.997 to 1.020
MELD	1.115	0.069	0.079	0.987 to 1.260
BCLC	1.003	0.363	0.994	0.493 to 2.039
GALAD	1.374	0.129	0.001	1.143 to 1.651
Treatment	3.907	3.121	0.088	0.817 to 18.694
**(B)**				
GALAD Score	1.285	0.071	<0.001	1.153 to 1.433

Abbreviations: AST, aspartate transferase; BCLC, Barcelona clinic liver cancer; GALAD, gender, age, AFP-L3, AFP, and des-carboxy-prothrombin; HR, hazard ratio; MELD, model for end-stage liver disease; se (HR), standard error of HR; 95% (C.I.), confidential interval at 95%.

## Data Availability

The original contributions presented in the study are included in the article. Further inquiries can be directed to the corresponding author.

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
