# Peer review of "Galad Score as a Prognostic Marker for Patients with Hepatocellular Carcinoma"

_ijms, 2023, doi:10.3390/ijms242216485_

Round 1

Reviewer 1 Report

Comments and Suggestions for Authors

I thank the editors for the opportunity to review the manuscript by Cagnin et al. The authors present an analysis of the Galad score as a prognostic marker for patients with HCC. All in all, the colleagues present an interesting work with a strong methodological background. The scientific question is current, and the underlying cohort has sufficient numbers. I have minor points, that I would like to address:

- The introduction is quite long. I recommend to shorten the section by half.

- Please give a reference for the calculation of the Galad score (p. 10, l. 328).

- I wonder if it would be possible to refine the stratification of the Galad score in more groups than two, and present its impact on survival.

Comments on the Quality of English Language

Overall, language, grammar and spelling are fine. Only minor edits need to be made, for example: p. 1, ll 26 ff: "Biomarkers were higher"

Author Response

Revisions have been attached

Reviewer 2 Report

Comments and Suggestions for Authors

The work submitted for review raises the important issue of diagnosis of hepatocellular carcinoma (HCC) using the GALAD score. Based on the GALAD score, the probability of HCC can be further calculated. These well-established HCC risk estimation algorithms were developed by Johnson et al. in 2014 (doi: 10.1158/1055-9965.EPI-13-0870).

The authors stated that the purpose of this study was to move from the current use of the GALAD score as a screening tool for early stage disease to its possible use as a prognostic score.

I did not notice whether the authors cite a recently published review on this topic:(Guan MC, Zhang SY, Ding Q, Li N, Fu TT, Zhang GX, He QQ, Shen F, Yang T, Zhu H. The Performance of GALAD Score to diagnosis of hepatocellular carcinoma in patients with chronic liver diseases: a systematic review and meta-analysis. J Clin Med. 26 Jan 2023;12(3):949. doi: 10.3390/jcm12030949. ), if not available, please cite this work.

In their study, the authors confirmed the diagnostic effectiveness of the GALAD scale.

They suggest authors the following points to consider before publishing:

1. justification should be added at the end of the introduction:

-what is the novelty of the research undertaken, why did the authors undertake this research, why is it important?

2. the editing of the work should be improved

3. the readability of drawings should be improved.

4. The conclusion of the work is too general and obvious

Author Response

Revisions have been attached

Round 2

Reviewer 2 Report

Comments and Suggestions for Authors

The manuscript has been improved and it could be published as it is.

Author Response

Thank you for your opinion.